# Enhancing Autonomous Driving in Urban Scenarios: A Hybrid Approach with Reinforcement Learning and Classical Control

**DOI:** 10.3390/s25010117

**Published:** 2024-12-27

**Authors:** Rodrigo Gutiérrez-Moreno, Rafael Barea, Elena López-Guillén, Felipe Arango, Fabio Sánchez-García, Luis M. Bergasa

**Affiliations:** Electronics Departament, University of Alcalá (UAH), 28805 Alcalá de Henares, Madrid, Spain; rodrigo.gutierrez@uah.es (R.G.-M.); rafael.barea@uah.es (R.B.); elena.lopezg@uah.es (E.L.-G.); juanfelipe.arango@uah.es (F.A.); fabio.sanchezg@uah.es (F.S.-G.)

**Keywords:** autonomous driving, deep reinforcement learning, decision-making, vehicle control, CARLA simulator

## Abstract

The use of Deep Learning algorithms in the domain of Decision Making for Autonomous Vehicles has garnered significant attention in the literature in recent years, showcasing considerable potential. Nevertheless, most of the solutions proposed by the scientific community encounter difficulties in real-world applications. This paper aims to provide a realistic implementation of a hybrid Decision Making module in an Autonomous Driving stack, integrating the learning capabilities from the experience of Deep Reinforcement Learning algorithms and the reliability of classical methodologies. Our Decision Making system is in charge of generating steering and velocity signals using the HD map information and sensors pre-processed data. This work encompasses the implementation of concatenated scenarios in simulated environments, and the integration of Autonomous Driving modules. Specifically, the authors address the Decision Making problem by employing a Partially Observable Markov Decision Process formulation and offer a solution through the use of Deep Reinforcement Learning algorithms. Furthermore, an additional control module to execute the decisions in a safe and comfortable way through a hybrid architecture is presented. The proposed architecture is validated in the CARLA simulator by navigating through multiple concatenated scenarios, outperforming the CARLA Autopilot in terms of completion time, while ensuring both safety and comfort.

## 1. Introduction

The rapid progression of Autonomous Driving (AD) has made significant inroads into both industry and academia. It has evolved into one of the most intensively researched fields, experiencing exponential growth in recent years [1]. Despite this, most vehicles currently include only Advanced Driver Assistance Systems (ADAS) as a pathway to achieve full AD. However, the ongoing advancements in AD, coupled with an analysis conducted by McKinsey [2], indicate that ADAS and AD combined could potentially contribute a substantial USD 300 billion to USD 400 billion to the passenger car market by the year 2035.This insight underscores the profound significance of AD in shaping the future of the automotive industry. Without a doubt, these improvements in Autonomous Vehicles (AVs) promise to reduce the number of accidents on the road. Over a million people die in traffic-related accidents each year. By removing human factors from vehicle control, AD could significantly enhance traffic safety [3].

In the field of AD, Decision Making (DM) is undoubtedly one of the most critical aspects influencing the vehicle’s behaviour. The ability to determine the appropriate action based on the surrounding environment, in a manner that is both safe and efficient, is a fundamental concern in this domain [4]. It is universally recognised that every AD system incorporates a DM module, aimed at minimising human errors in driving tasks. This research is centred on these crucial aspects, to develop a framework for autonomous DM in realistic scenarios within the AD context.

Deep Reinforcement Learning (DRL), a subset of Machine Learning (ML), emerges as a promising candidate to help in the decision task [5]. As delineated above, AD challenges within DM offer a clear avenue for exploration. DRL appears to be an ideal tool for the task, given its close alignment with the Partially Observable Markov Decision Process (POMDP), its adaptability to diverse domains, and its applications in DM engineering. At its core, DRL combines Reinforcement Learning (RL) with Deep Neural Networks (DNNs). By integrating the principles of Bellman’s equation with the advanced capabilities of Neural Networks (NNs), DRL can analyse environments and infer the optimal decision for a given set of inputs, surpassing the capabilities inherent to traditional RL. This process occurs through repeated trial and error experiments as humans do, during which an agent incrementally acquires a desired behaviour by continuously interacting with its environment.

However, the application of DRL techniques does not always satisfy the requirements of AD. As mentioned before, the main problem to be solved by AVs is related to traffic safety, where DRL approaches may have some difficulties due to their primary focus on optimising a task without considering comfort and safety. These techniques are divided into model-based and model-free. Model-based methods incorporate vehicle dynamics but frequently encounter concerns, such as high computational demands and the complexity of creating accurate environmental models, which compromise reliability in unpredictable scenarios like those found in AVs [6]. On the other hand, model-free methods seek to find the best solution to a specific problem, without taking into account the vehicle’s dynamics. By a trial and error process, these methods learn a certain behaviour, which is not usually the safest nor the most comfortable, but that identifies high-level behaviours. This is why we propose a hybrid approach in this work that mixes the best of the DRL and classical control approaches. While a model-free DRL algorithm takes care of choosing the optimal high-level actions, some other classic modules are in charge of safe and comfortable movements defined for the actions, including the vehicle’s dynamics.

This work develops a hybrid hierarchical DM architecture which generates steering and velocity signals for navigating in urban scenarios. The adoption of hierarchical systems is widely acknowledged in the academic community [7]; however, unlike hierarchical DRL [8] that presents practical implementation concerns, our proposal uses a classical control system for generating vehicle movement signals, which significantly improves driving smoothness and safety. This makes it possible to develop a realistic approach for driving in various complex urban scenarios, transcending the typical focus on singular use cases found in the existing literature based on RL. The key contributions of the proposal are summarised as follows:Contribution 1: This paper proposes a hybrid methodology that integrates multiple components: pre-processing of map information, high-level DM facilitated by the DRL module, and low-level control signals managed by a classic controller. Our approach not only solves individual complex urban scenarios but also handles concatenated scenarios. This contribution is an extension of the work previously published in the conference IV 2023 [9].Contribution 2: In this work, a novel low-level controller is developed. This includes a Linear–quadratic regulator (LQR) controller for trajectory tracking and a Model Predictive Control (MPC) controller for manoeuvre execution. The online integration of these two controllers results in a hybrid low-level control module that allows for the execution of high-level actions in a comfortable and safe manner.Contribution 3: This study also presents a RL framework developed within the Car Learning to Act (CARLA) simulator [10] to evaluate complete vehicle navigation with dynamics. Unique to this framework is the incorporation of evaluation metrics that extend beyond mere success rates to include the smoothness and comfort of the agent’s trajectory.

## 2. Related Works

The DM module in AD essentially acts as the operational “brain” of the vehicle. This module plays a pivotal role in ensuring both safe navigation and operational efficiency. Many studies in the literature concentrate on specific DM tasks [9,11]; however, our research encompasses the entire AD stack. In this section, we review some representative AD architectures in the literature, particularly in the context of DM and DRL.

### 2.1. Transformer-Based Reinforcement Learning

The use of transformer-based algorithms has been gaining relevance in recent years in the context of AD [12], with various works following this approach [13,14]. One of the most relevant and representative frameworks in this line employs a Scene-Rep Transformer to enhance RL DM capabilities [15]. This method allows for the system to function independently of scenario-specific implementations, covering a range of scenarios with a singular approach. The selected state includes the past trajectories of the vehicles within the scenario and the future potential centrelines for each vehicle. The actions proposed in this work are the longitudinal velocity of the ego vehicle and a lane change signal, executed by the SUMO simulator. Experiments are carried out in both hyper-realistic and interactive driving environments. This work addresses three distinct urban scenarios independently but fails to offer a framework capable of navigating through these scenarios in a concatenated manner.

### 2.2. Attention-Based Deep Reinforcement Learning

Another trend is attention-based architectures [16,17,18]. Specifically, the authors of [18] propose an attention-based driving policy combined with an LSTM network for the estimation of the value function. The aim is to manage unprotected intersections, employing DRL. This proposal places a greater emphasis on realistic implementation within an AD architecture. They utilise past trajectories and discrete route information as the state input and use a target velocity for the action space. A PID controller is responsible for generating the throttle and brake signals from this target velocity. In these scenarios, trajectory tracking is managed by an external controller, which implies that steer-related actions are not generated by the DRL agent. Lane changing is not considered; thus, decisions are limited to adjusting the longitudinal velocity. This study demonstrates good results in terms of success rate primarily focusing on various intersection scenarios. However, velocity profiles, indicate the presence of abrupt control commands, leading to significant fluctuations in the target velocities.

### 2.3. Combining Machine Learning and Rule-Based Algorithms

Some other works propose a hybrid strategy that combines the advantages of rule-based and learning-based methods for DM and control, aiming to mitigate their limitations [19,20]. The authors in [21] present a practical implementation, with emphasis on trajectory generation. This work opts for a simple but effective implementation. The authors propose a low-dimensional state space, with low-level controllers in charge of generating the driving commands. The graphical representation of this work shows smooth control signals, which are easily followed by a model with dynamics. However, the work lacks diversity of scenarios, focusing mainly on a single environment in simulation.

### 2.4. Tactical Behaviour Planning

Other works focus on solving various scenarios using a tactical behaviour planning [22,23]. The authors of [24] propose high-level DM for AD. They formulate the problem as a continuous POMDP, combining the advantages of two State of the Art (SOTA) solvers Monte Carlo Value Iteration and Successive Approximations of the Reachable Space under Optimal Policies. The authors also focus on high-level actions concerning linear velocity, specifically addressing throttle and brake commands, to which the vehicle’s dynamic model responds. The operational behaviour of the proposed framework demonstrates how the vehicle follows the control commands. However, in the presented results, there is a notable lack of comfort metrics, with only velocity being considered. This approach shows effective behaviour across various scenarios, but the utilisation of these algorithms presents challenges such as high computational costs and scalability issues.

### 2.5. Discussion

The comparative analysis presented in Table 1 delineates the distinctive attributes and efficiencies of the previously introduced DRL approaches, including Transformer-based, Attention-based, Decision-Control, Tactical Behaviour, next to our proposed methodology. This discussion highlights the distinct features and evaluates the overall effectiveness and feasibility of these architectures in the AD context regarding the following parameters.

State Dimensionality and Pre-Processing: The architectures employing Transformer and Attention mechanisms are characterised by handling high state dimensionalities, utilising sophisticated pre-processing techniques to manage complex input data. Conversely, the Decision-Control, Tactical Behaviour, and our approach, with an emphasis on low state dimensionalities, leverage simpler pre-processing methods such as map data, aiming for computational efficiency and reduced complexity in data handling.Action and Control Signal: Transformer-based and Attention-based models provide low-level control commands, often resulting in sharper vehicle control. In contrast, our methodology, along with Decision Control and Tactical Behaviour, opt for high-level actions that yield smoother control signals, promoting more naturalistic and comfortable driving behaviours.Scenario Handling: The capacity to adapt to multiple, including concatenated, scenarios shows the versatility of DRL models in AD. Our approach, similar to Transformer-based and Attention-based models, supports a broad spectrum of driving situations, crucial for developing adaptable and dynamic AD systems.Computational Cost and Scalability: In terms of computational cost, both our proposal and Tactical Behaviour have lower costs and higher efficiency. Moreover, our model, together with the Transformer-based and the Attention-based models, show scalability, being easily transferable from one environment to another.Real Implementation: Among the reviewed architectures, only our approach and the Attention-based model have been validated in real-world settings, showcasing their reliability and applicability beyond simulated environments.

In conclusion, key to our methodology is the generation of smooth control signals, which are evaluated against comfort metrics to ensure a comfortable and safe driving experience. Unlike many existing systems that struggle with abrupt or jerky movements, our system prioritises the smoothness of manoeuvres, making it more aligned with human driving behaviours. Our system stands out by handling concatenated driving scenarios, showing its flexibility and ability to adapt to different driving situations. Another advantage of our approach is its operation within a low-dimensional state space, which, coupled with its low computational cost, makes it an efficient and scalable solution for AD.

## 3. Background

This section delves into the foundational concepts and methodologies underpinning DM in AD. First, we provide an overview of the AD landscape. Moving forward, we introduce the mathematical framework used for our DM implementation. Finally, we present the algorithm used in our proposal.

### 3.1. Autonomous Driving

Current AD SOTA identifies four principal software architecture categories [25]: end-to-end, modular, ego-only, and connected. End-to-end architectures are seen as “black-box” models where a single Neural Network (NN) manages the entire driving task directly from raw sensor data. This could potentially eliminate errors as intermediate representations are optimised. However, these models suffer from a lack of interpretability and present challenges in their implementation within real systems [26,27]. Conversely, modular architectures, often referred to as “glass models”, decompose the driving task into separate modules, independently programmed or trained. Such architectures enhance interpretability, facilitate knowledge transfer, and allow for parallel development, making them a standard in industrial research. However, they carry the risk of error propagation through the intermediate stages, potentially leading to suboptimal performance. The integration of these AD stacks is frequently mentioned in the literature as a common approach for real-world applications [25,28]. The ego-only approach involves performing all necessary automated driving operations independently within a single self-sufficient vehicle at all times. Conversely, connected systems utilise Vehicle-to-Everything (V2X) communication to exchange information with other vehicles, infrastructure, and centralised servers.

Table 2 provides a comparison of the four approaches, summarising their advantages and disadvantages.

The hybrid DM architecture proposed in this work is integrated into the AD stack of the Robesafe group [29]. Figure 1 illustrates the modular approach adopted by the group which consists of the following modules: perception, localisation, mapping, planning, decision, and control; and identifies the main modules developed in this paper (outlined with red dashed lines).

This work is focused on the decision module, which is a critical component in the architecture of AVs. It serves as the brain of the vehicle, processing information from various sensors and subsystems to make informed decisions that ensure safe and efficient navigation. This module’s primary responsibility is to interpret the environment, predict the actions of other road users, and determine the best course of action in real time, making it indispensable for the successful operation of self-driving vehicles. There are several approaches to develop the decision module in AD [30]:Rule-Based Systems: Use a predefined set of rules to guide the vehicle’s decisions.Finite State Machines: Model the DM process as a series of states and transitions.Behaviour Trees: This approach structures the DM process in a tree-like hierarchy, allowing for modular and scalable systems.Machine Learning and Deep Learning: These techniques, where DL is a subset of ML, enable vehicles to learn from data and make decisions based on features extracted from data.Reinforcement Learning: is a subset of ML, characterised by its focus on training an agent through experiments to interact effectively with its environment.

Among the various approaches, RL stands out as a promising method for tackling the challenges of DM in uncertain and complex environments, making it an attractive option for the future of AD technology.

### 3.2. Partially Observable Markov Decision Processes

Partially Observable Markov Decision Processes (POMDPs) are a mathematical framework for the RL implementation. At its core, a POMDP extends the Markov Decision Process (MDP) framework by incorporating elements of uncertainty in the observation of the system’s state [31]. A POMDP can be formally defined by the tuple (S,A,T,R,Ω,O), where:*S* is a finite set of states, representing the possible configurations of the environment.*A* is a finite set of actions available to the decision-maker or agent.T:S×A×S→[0,1] is the state transition probability function and $T(s,a,s^{\prime })$ represents the probability of transitioning to state $s^{\prime }$ from state *s* after taking action *a*.R:S×A→R is the reward function, associating a numerical reward (or cost) with each action taken in a given state.Ω is a finite set of observations that the agent can perceive.O:S×A×Ω→[0,1] is the observation function, $O(a,s^{\prime },o)$ defines the probability of observing *o* after taking action *a* and ending up in state $s^{\prime }$.

Due to partial observability, the agent cannot directly access the true state of the environment. Instead, it maintains a belief state, a probability distribution over the set of possible states, representing its degree of certainty about the environment’s actual state. The agent updates this belief state based on its past actions and observations.

### 3.3. Deep Reinforcement Learning

In the context of MDPs, the state is fully observable and the Markov property implies that the future state is dependent only on the current state and action. However, in POMDPs, the state *s* of the system is not fully observable, so the agent receives observations *o* that provide partial information about the state. This implies that, while the underlying process is Markovian (the next state depends only on the current state and the action taken), the agent does not have full visibility of the state. This is the case of our DM, which takes into account the current belief state to make decisions. Due to this, POMDPs require a different approach compared to standard MDP for determining this belief state and learning value functions or policies. In DRL, the agent effectively determines the belief state and learns both the value function and the policy without the need for explicit feature engineering, leveraging the capabilities of DNNs. In this work, we implement an approach wherein DNNs are used to learn the value function, policy, and internal state of the agent modelled as a POMDPs in charge of the DM of our AD stack.

With numerous DRL algorithms available, we selected the Trust Region Policy Optimization
(TRPO) [32] algorithm to be integrated within our proposal. An ablation study on various DRL algorithms, including DQN, A2C, TRPO, and PPO, was conducted in our work [33]. The results of the study demonstrate the superiority of the TRPO algorithm over others in various scenarios.

TRPO optimises policy parameters while ensuring monotonic improvement in policy performance. The key idea in TRPO is to take the largest possible step in the policy space without causing a significant deviation in the behaviour of the policy. This is achieved by optimising a surrogate objective function subject to a trust region constraint. To ensure that the new policy is not too far from the old policy, a constraint based on the Kullback–Leibler divergence is used:(1)$$\mathrm{KL}[\pi_{\theta_{\mathrm{old}}}\left(\cdot |s\right),\pi_{\theta }\left(\cdot |s\right)]\leq \delta$$
where δ is a small positive number that defines the size of the trust region.

The loss function in TRPO is defined as follows:(2)$$L\left(\theta \right)={\hat{E}}_{t}\left[\frac{\pi_{\theta }\left(a_{t}|s_{t}\right)}{\pi_{\theta_{\mathrm{old}}}\left(a_{t}|s_{t}\right)}{\hat{A}}_{t}\right]$$

Here, $\pi_{\theta }\left(a_{t}|s_{t}\right)$ represents the probability under the new policy parameters θ of taking action $a_{t}$ given state $s_{t}$, $\pi_{\theta_{\mathrm{old}}}\left(a_{t}|s_{t}\right)$ is the probability under the old policy parameters, and ${\hat{A}}_{t}$ is an estimator of the advantage function at time *t*.

## 4. Methodology

This section provides an overview of the proposed hybrid DM architecture, focusing on its general structure and delving into each module of the AD stack, highlighting how these modules interact. As elaborated in the previous section, the current SOTA showcases various research directions, including the application of DL-based methods, the utilisation of traditional algorithms, and the formulation of hybrid approaches. Our architecture has been inspired by and developed in line with these hybrid proposals. In this way, this work tackles the integration of different modules in an AD stack, encompassing high-level DM and local manoeuvre control within a hybrid architecture. Our system is uniquely designed to interact with both simulated and real-world environments. It has three primary inputs: vehicle location, HD map information, and sensor data.

The architecture of our system is structured into four levels, as illustrated in Figure 2. The perception level is responsible for processing sensor data and generating the global location and velocity of the surrounding vehicles. The decision module of an AD stack is typically divided into three tasks: global, local, and behavioural planning. The global planning is executed by the strategy level, where a tactical trajectory is defined based on the HD map information. This level lays the foundation for navigation and routing. Behavioural planning is carried out by the tactical level, corresponding to our high-level DM module. Here, high-level decisions are made by the DRL agents, guiding the vehicle’s behaviour across various driving scenarios. Local planning and low-level control are undertaken by the Manoeuvre Execution module and the Trajectory Tracking module, respectively. These modules constitute the operative level, translating high-level decisions into actionable control commands and managing the vehicle’s movements and interactions with its environment. This work contributes mainly to the tactical (DM module) and operative (local planning and control module) levels, with a small contribution at the strategy level (global planning module).

Overall, this approach allows for a clear division of functionalities within the AD stack, facilitating effective and efficient AD operations. In the following sections, we will describe the levels of the proposed architecture.

### 4.1. Strategy Level

Within the strategy level, two modules are found: the global planner and the scenario planner. The global planner is tasked with defining the global trajectory, which generates the route the vehicle has to follow. In contrast, the scenario planner generates the tactical trajectory, containing the locations of different driving scenarios on the map. The architecture of the strategy level is presented in Figure 3.

The strategy level is based on prior research presented in [34], where the global planner module was developed. For this work, we have expanded upon their work by developing the scenario planner. The global planner receives the ego-vehicle’s position and goal position and calculates the optimal route between them as a list of waypoints centred in the lanes using the Dijkstra algorithm [34]. A waypoint is a structured object representing a 3D point with location (x, y, z), rotation (pitch, yaw, roll), and additional topological information from the HD map, including road and lane ID, lane width, lane markings, and the speed limit of the road. This module comprises two planners. The first planner, known as the Lane Graph Planner (LGP), establishes a topological (road-lane) route. Following this, the Lane Waypoint Planner (LWP) takes charge of generating waypoints. These waypoints collectively form the global trajectory.

The map information encapsulated within the waypoints is used by the scenario planner to create the tactical trajectory. This tactical trajectory delineates the start and end points of various driving scenarios along the route. It serves as a crucial input for a high-level selector. This selector is a ruled-based system which, by considering this trajectory and the vehicle’s current location, determines whether the vehicle is within a specific scenario (e.g., roundabout, merge, etc.).

In summary, this level uses three inputs: the current location of the vehicle, comprehensive map data, and a destination point. From these inputs, the system delineates two critical trajectories:Global Trajectory: Consists of a sequence of waypoints that define the path to be followed by the vehicle from its starting point to the destination.Tactical Trajectory: Composed of scenario-specific waypoints, strategically positioned within the map. These waypoints mark the start and end of scenarios encountered along the route.

### 4.2. Tactical Level

The tactical level is responsible for processing and evaluating information received from other levels to make decisions. The inputs for this level include the locations of different scenarios, which are provided in the tactical trajectory, HD map data, and the perception output (the actual vehicle location and the location and velocities of adversarial vehicles). The output of the tactical level is a high-level action, which essentially dictates the vehicle’s immediate behaviour. These decisions are stopping, continuing to drive, and executing lane changes, which are then executed by the operative level. An overview of the tactical level is illustrated in Figure 4.

The tactical level encompasses three modules: scenario selector, perception data pre-processing, and behaviour modules, commonly referred to as agents.

Scenario Selector: This module is in charge of selecting the agent to be executed. In the tactical trajectory, the locations where each use case starts and ends are stored. The “Follow Path” agent is activated by default. When the vehicle reaches one of these locations, the selector activates the corresponding agent. Once the end of the use case is reached, the “Follow Path” agent is activated again.Pre-Processing: Another task is the pre-processing of perception data, which involves transforming global locations and velocities of surrounding vehicles into an observation vector. This process starts by obtaining the global location of each vehicle, which is then mapped to a specific waypoint on the HD map. Subsequently, each waypoint is associated with a particular lane and road. This information, coupled with the current scenario as determined by the Scenario Selector, forms the basis for generating distinct observation vectors. Importantly, these vectors are scenario-specific, varying according to the different driving situations encountered.DRL Agents: In the proposed architecture, five distinct behaviours (use cases) can be executed. By default, the "Follow Path" behaviour is selected, where the operative level follows the global trajectory while maintaining a safe distance from the leading vehicle, and no active decisions are made. Upon the Scenario Selector choosing a specific scenario, one of the following agents is activated (Lane Change, Roundabout, Merge, Crossroad). These agents then take actions (drive, stop, turn left, turn right) based on the corresponding observation vector.

### 4.3. Operative Level

The operative level is responsible for two primary tasks: following the global trajectory generated by the strategy level (Trajectory Tracking module), and executing high-level actions determined by the tactical level (Manoeuvre Execution module).

For a smooth trajectory, spline interpolation is employed, offering significant advantages in computational efficiency, accuracy, and smoothness over alternatives such as clothoids [35] and polynomial curves [36,37]. Spline curves [38,39] are particularly advantageous due to their low computational cost and high degree of smoothness at segment joins. Once the trajectory is defined, a lateral controller provides closed-loop path tracking. Among tracking controllers, the Pure Pursuit [40] and Stanley [41] controllers are widely used benchmarks but suffer from limitations like parameter tuning and dynamic feedback requirements. Our approach employs an LQR controller with delay compensation, enabling stable and efficient real-time execution at medium driving velocities [42].

For manoeuvres, MPC methods have been implemented to incorporate lateral and longitudinal constraints, ensuring smooth transitions during manoeuvres. While sampling-based [43] and Optimal Control Problem methods [44] are common approaches, they often face challenges in dynamic environments. The need to frequently update the trajectory, particularly with MPC [45,46], imposes significant computational demands [47]. We integrate an online MPC controller enabling real-time adjustments to the spline-based nominal trajectory. This approach ensures both computational efficiency and reliable solutions, addressing the limitations of existing techniques. A deeper explanation was published by the authors in [48].

The architecture of the operative level is illustrated in Figure 5, which highlights the interaction among the two controllers within this level. In “Follow Path” behaviour, the vehicle follows the nominal commands in a smooth trajectory set by the Trajectory Tracking module. When an action is selected by the Tactical Level, the Manoeuvre Execution module modifies the nominal control signals in a smooth way. This ensures that the desired action is executed smoothly and comfortably.

Trajectory Tracking: Utilising the provided waypoints, a smooth trajectory is computed using the LQR controller. At each simulation time step, the lateral and orientation errors are calculated to generate a steering command aimed at minimising these errors. Moreover, a velocity command is derived based on the curvature radius of the trajectory. These steering and velocity commands constitute the nominal commands, enabling the vehicle to follow the predefined path accurately.Manoeuvres: In the operative level, manoeuvres modify the nominal commands based on the tactical level’s requirements, covering three primary tasks. Firstly, when a preceding vehicle is detected, the MPC controller adjusts the ego-vehicle’s velocity to adapt it to the ahead-vehicle velocity keeping a safe distance. Secondly, if a stop action is demanded by an agent, the decision velocity decreases smoothly, being the resulting command the minimum between this constrained velocity and the nominal velocity. Lastly, in the case of a lane change request (left or right), the MPC controller generates a lateral offset to modify the vehicle’s location, and the LQR controller generates a smooth steering signal to facilitate the lane change.

## 5. Experiments

Urban driving environments encompass a variety of scenarios. In this work, we identify and explore four key scenarios common in many cities: crossroads, merges, roundabouts, and lane changes. We formulate these scenarios using POMDPs, treating each scenario independently. This method allows for a segmented understanding of the DM process, breaking it down into distinct tasks. The vehicle under control, referred to as the “ego vehicle”, is defined as an agent. This agent gathers data from the environment in the form of observations and executes actions based on a defined policy. This policy updates through the training process, which utilises the reward function.

The TRPO algorithm in this study is implemented using the SB3 library [49]. An agent is trained for each use case over one million time-steps, ensuring convergence. The NN architecture is divided into two main components: a features extractor module and the DRL algorithm.

Features Extractor Module. In line with insights from our previous research, this work incorporates a feature extraction module, which has proven to enhance the convergence of training [50], comprising a dense Multi-Layer Perceptron (MLP) to process observations from the environment. Information about both adversarial and ego vehicles is separately processed through the feature extractor. The outputs are then concatenated into a single vector, serving as the input for the DRL algorithms.Deep Reinforcement Learning Algorithms. An actor–critic framework is adopted. Within this framework, one MLP functions as the actor, determining the actions to take, while a separate MLP serves as the critic, evaluating the action’s value.

All MLP present two hidden layers, with each layer comprising 128 neurons, and utilise the *tanh* activation function. The input layer’s dimension is based on the number of elements in the observation vector. The dimension of the output layer corresponds to the number of possible actions. A representation of the framework is depicted in Figure 6.

### 5.1. POMDP Formulation

The POMDP formulation for the four use cases is characterised by low-dimensional representations. However, there are differences in the number of vehicles and types of actions depending on the scenario.

#### 5.1.1. State

The state of a vehicle is defined by its distance to a relevant point (shown as $d_{i}$ in Figure 7), its longitudinal velocity, and its driving intention: $s_{i}=\left(d_{i},v_{i},i_{i}\right)$. These values are normalised between [0,1] and the driving intention of the adversarial vehicles are described by i∈[0,2]. In the lane change scenario, a vehicle may have three intentions: change left (i=1), keep driving in its lane (i=0), or change right (i=2). In the crossroad scenario, these intentions are related to the trajectory to be followed: turn left (i=1), continue straight (i=0), or turn right (i=2). Regarding the merge scenario, the vehicles may yield (i=0) or take the way (i=1). Finally, in the roundabout scenarios, the intentions are to leave before (i=0) or after (i=1) the ego vehicle exits. We define the state of the environment as the collection of the individual states of the adversarial vehicles: $s=(s_{1},s_{2},...,s_{n})$, being *n* the number of adversaries within the scenario.

#### 5.1.2. Observation

The observation matrix is defined by the nearest vehicles to the ego vehicle, as shown in Figure 7. The observation space is defined by two key components: the relative normalised distances to surrounding vehicles, for the lane change scenario, and the intersection point, for the rest of the scenarios, and the normalised velocities of these vehicles. This approach ensures a comprehensive and scaled representation of the vehicle dynamics and spatial relationships in the system.

#### 5.1.3. Action

The agent only controls the high-level decisions while the velocity is automatically controlled by the operative level. We propose a discrete set of actions: change to the left lane, continue in the current lane, and change to the right lane for the lane change scenario. Alternatively, for the rest of the scenarios, the possible actions are “drive” and “stop”.

#### 5.1.4. Reward

We aim to drive at a desired velocity without colliding with adversarial vehicles. A positive reward is given when the vehicle reaches the end of the road and a negative reward is given when it collides. We propose a small cumulative reward based on its longitudinal velocity, to ensure that the vehicle tries to drive as fast as possible. The values of each component of the reward function have been chosen after testing different approaches. The reward function is defined as follows.
Reward based on the velocity: $k_{v}*v_{ego}$;Reward for reaching the end of the road: +1;Penalty for collisions: −2.
where $k_{v}=1\times 10^{-3}$. Under this setup, the episode reward is in the range of [−2, 1.1], due to the small values of the constants for velocity and right lane adherence. This range allows us to measure the vehicle’s performance based on the average reward per episode.

### 5.2. Evaluation Metrics

In order to comprehensively evaluate the performance of an RL agent, a thorough analysis is conducted focusing on safety, comfort, and efficiency metrics. Our evaluation is divided in two parts: a quantitative analysis and a qualitative analysis. Firstly, we examine various numerical metrics, each of which offers insights into different aspects of an agent performance:Success Rate (%): This metric indicates the frequency of succeed episodes performed by the agent during simulation, providing a direct measure of safety.Average of 95th Percentile of Jerk (per episode, in m/s^3^): Jerk is the rate of acceleration changes. This metric reflects the smoothness of the driving, relating to passenger comfort.Average of Maximum Jerk (per episode, in m/s^3^): This metric measures the highest jerk experienced.Average of 95th Percentile of Acceleration (per episode, in m/s^2^): This metric provides insight into how aggressively the vehicle accelerates, impacting both comfort and efficiency.Average Time of Episode Completion (in seconds): It measures the duration taken to complete an episode, indicating the efficiency of the agent.Average Speed (in m/s): This metric assesses the agent’s ability to maintain a consistent and efficient speed throughout the episode.

### 5.3. Concatenated Use Cases Scenario

In our setup, the perception module directly obtains ground truth information from the simulator, bypassing the need for real-world sensor data processing. The scenario is set in the Town03 map, selected for its diverse use cases and high level of realism. We develop a scenario wherein the ego vehicle navigates through diverse urban environments under varying driving conditions. To achieve this, we ensure a diversity of routes and behaviours, as well as sufficient complexity in terms of traffic density and vehicle dynamics. We outline two steps to define a scenario. Initially, an ideal spot is identified, where a sufficient number of vehicles can navigate. There are four concatenated scenarios where the adversarial vehicles spawn every 3 to 5 s: the Lane Change scenario where vehicles may change lanes or continue straight, the Roundabout scenario with a 30-meter-radius roundabout, the Merge scenario which involves vehicles generated on a perpendicular lane, and the Crossroad scenario, where vehicles are generated at both sides of the intersection, proceeding straight or turning.

Subsequently, we define the vehicles’ routes and behaviours. This is accomplished using the TM integrated in PythonAPI 0.9.14, which enables us to randomly designate the vehicles’ trajectories. These vehicles can reach maximum speeds of 5 to 15 m/s (18 to 54 km/h) and their intentions match the ones introduced in Section 5.1. However, realistic simulators often encounter issues with computing times. To mitigate this problem, we employ synchronous simulation, where each simulation step is completed before the next begins. This ensures all tasks for a given step, like calculations and decisions, are fully processed within a fixed frame rate. Additionally, we ensure efficient traffic flow generation by spawning and destroying vehicles at critical points, thereby avoiding the presence of vehicles outside the scope of the scenario.

As depicted in Figure 8, the ego vehicle begins its route and approaches a roundabout, where it must safely enter. Upon exiting, it continues onto a two-lane road where vehicles in the right lane are moving slowly, needing an overtaking manoeuvre. Subsequently, it encounters a high traffic crossroad where it must identify a gap to cross. The final challenge involves merging right to complete the route. This scenario is available in our github (https://github.com/rodrigogutierrezm/carlagymrl, accessed on 15 December 2024).

### 5.4. Results

To evaluate the effectiveness (in terms of success rate) and efficiency (in terms of average completion times) of our approach, we conducted a comparative analysis against two established methods: the CARLA Autopilot and the modular AD stack baseline Techs4AgeCar architecture [29]. The results of our analysis are summarised in Table 3.

The CARLA Autopilot, which is the standard vehicle control method provided by the simulator, is governed by a PID controller and managed by the Traffic Manager (TM) module. This module ensures collision-free trajectories by coordinating the movements of all vehicles, including the ego vehicle, while making adversarial vehicles aware of its position and actions.

Conversely, the Techs4AgeCar architecture represents a modular approach, integrating a perception module for detecting surrounding vehicles, decision-making via Petri nets, and a low-level control system for generating vehicle commands. This architecture demonstrated its reliability by ranking second in the CARLA Autonomous Driving Leaderboard (Map Track), making it a robust baseline for comparison.

For this comparison, the maximum velocity of all three systems was set to 50 km/h to ensure fairness. While the Autopilot system boasts a flawless success rate of 100%, this outcome is significantly influenced by its privileged access to the simulation’s internal information. This insider advantage allows the Autopilot to navigate without the uncertainties that typically tackle autonomous systems. In contrast, our proposed system achieved an impressive success rate of 95.76%, outperforming the 92.84% achieved by the Techs4AgeCar proposal. These figures underline our system’s robustness and its adeptness at handling dynamic driving scenarios with limited information and the absence of data (such as the intentions of the adversarial vehicles and the locations and velocities of vehicles outside the sensor range). A deeper dive into the jerk dynamics reveals our system’s superior smoothness, with the 95th percentile of jerk per episode markedly lower that the others. Specifically, our system recorded jerk metrics of 1.87. This reduced jerk indicates a smoother driving, enhancing passenger comfort and safety. Furthermore, our system outperformed the other proposals in terms of manoeuvre completion time and average speed, evidencing its efficiency. Our system completed the manoeuvres in significantly less time (76.85 s) than the Autopilot (140.23 s) and the Techs4AgeCar stack (88.43 s). Moreover, our system maintained higher average speeds, showcasing its ability to navigate the environment not only more quickly but also more smoothly.

We showcase the temporal evolution of vehicle signals throughout the concatenated scenario, examining velocity, acceleration, jerk, and steering. The diagrams use different colours to represent the separate use cases, with Figure 9 illustrates the performance of our AD stack, while Figure 10 depicts the behavior of the CARLA Autopilot, and Figure 11 represents the Techs4AgeCar stack simulation.

The black lines in the diagrams denote periods outside specific scenarios, during which all methodologies begin with similar behavior. As the vehicles approach the roundabout, both our approach and the Techs4AgeCar system efficiently merge into the traffic flow. In contrast, the Autopilot system halts, unable to navigate through small gaps in traffic. Subsequently, all methodologies follow a leading vehicle; however, our approach ensures smoother transitions during the ACC manoeuvre. Notably, when the leading vehicle significantly reduces its speed, both our system and the Techs4AgeCar approach successfully execute overtaking maneuvers. In contrast, the Autopilot remains behind, resulting in jerk signal spikes caused by alternating acceleration and braking. Similar trends are observed in the crossroad and merge scenarios. Our system demonstrates rapid and smooth actions, while both the Autopilot and Techs4AgeCar exhibit erratic behavior, struggling to merge properly into traffic.

Overall, the graphical analysis underscores our methodology’s superior velocity and smoothness, attributable to the integration of DRL modules for scenario complexity management and classical low-level control for ensuring safety and comfortability. Concretely, the MPC ensures that the longitudinal and lateral jerk do not surpass a certain threshold, maintaining passenger comfort, while the LQR guarantees smooth trajectory tracking, avoiding discontinuities.

### 5.5. Influence of Realistic Sensors

The main advantage of working with a hyper-realistic simulator is the possibility of using realistic sensor models. This experimental section ends with the study of the influence of virtual sensors in the DRL agent regarding an agent trained using only ground truth data. For this experiment, we have not added another training stage, so we can analyse the effects of the uncertainty produced by the sensor’s detection. We present a first approach, where the observation matrix is directly obtained from the CARLA simulator; and a second realistic approach, where this matrix is defined using a detection module based on LiDAR with PointPillars [51] and the HD map information provided by the simulator.

In Table 4, we present the evaluation metrics of these approaches, where the results obtained become worse with the sensors data, appreciating a deterioration in the performance with adverse weather conditions.

## 6. Conclusions and Future Works

We developed an innovative hybrid DM architecture for AD systems, integrating classical control techniques with DRL. This hybrid approach combines the reliability of traditional control methods with the adaptive capabilities of DRL to changing scenarios. The proposed architecture is currently integrated within an AD stack, but its design is sufficiently flexible to be adapted to other architectures. The proposed architecture has surpassed traditional RL approaches by effectively deploying DRL algorithms in a hyperrealistic simulation, successfully solving various concatenated urban scenarios. Despite these advances, our proposal has some limitations, particularly in adapting to new situations and scenarios. This challenge is related to both the algorithms employed and the scenario generation process.

The focus of this work proposal lies in a classical implementation of DRL, as the aim is to integrate it into a complete AD architecture. However, there are several lines of research within RL that can improve the presented architecture and solve the limitations found. This includes transfer learning, where previously trained models can be used to enhance our approach; end-to-end methods, which allow the system to adapt more easily to new scenarios by extracting the state vector directly from sensor data; and Inverse Reinforcement Learning, which can learn a policy based on expert knowledge, resulting in more naturalistic behaviour. Regarding simulation, countless possibilities exist wherein we identify lines of research, such as scaling our proposal by incorporating more diverse traffic situations or creating a framework for the automatic generation of scenarios were the adversarial vehicles present a human-like behaviour.

The ultimate goal of this architecture is to be deployed on a real platform. A more in-depth analysis of this issue is available in our work [33].

## Figures and Tables

**Figure 1 sensors-25-00117-f001:**
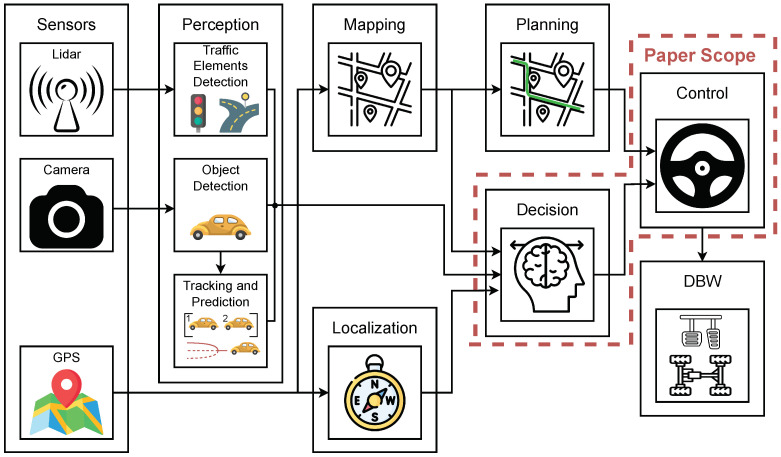
Autonomous driving stack modular pipeline.

**Figure 2 sensors-25-00117-f002:**
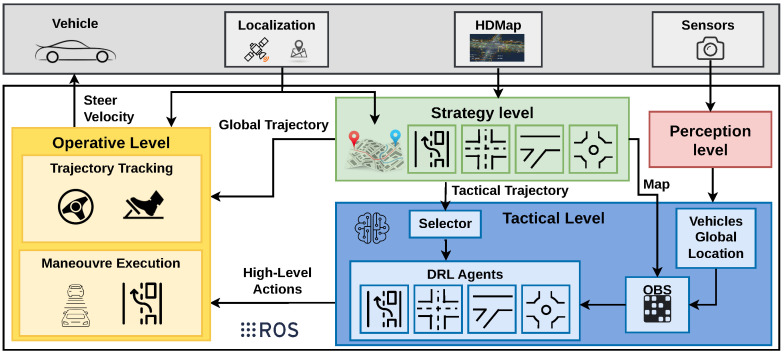
The proposed hybrid architecture. The strategy level defines a tactical trajectory with the map information and the ego vehicle location. The tactical level executes high-level actions in correlation with the perception ground truth. The operative level combines the trajectory and the actions, calculating the driving commands.

**Figure 3 sensors-25-00117-f003:**
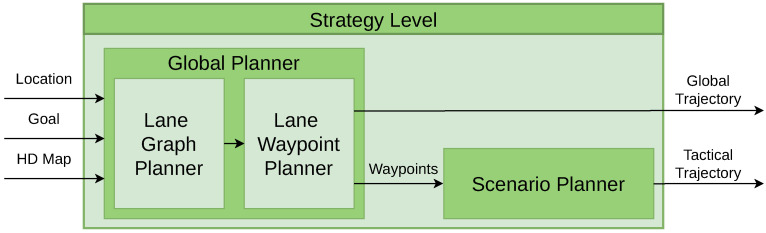
Strategy level. The global planner calculates the global trajectory while the scenario planner generates the tactical trajectory.

**Figure 4 sensors-25-00117-f004:**
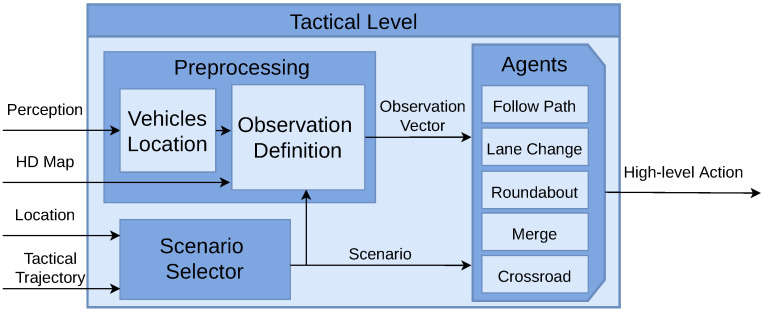
Tactical level. Perception data are processed in conjunction with the HD map to formulate the observation vector. The system selects a specific scenario based on the tactical trajectory and the current location. A decision module is selected, which is responsible of executing decisions.

**Figure 5 sensors-25-00117-f005:**
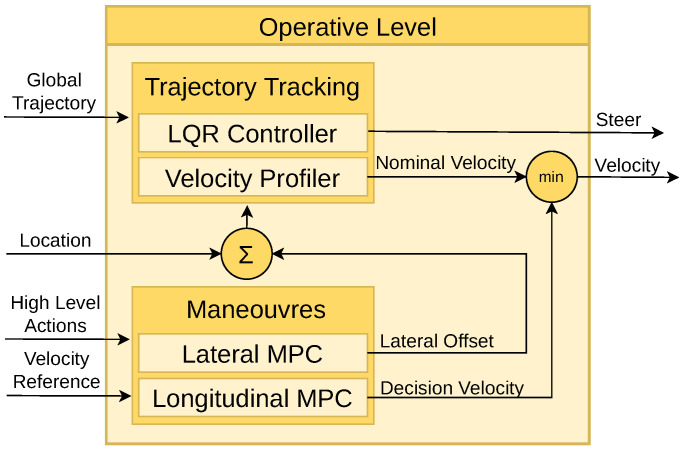
Operative level. The global trajectory is meticulously followed using a linear-quadratic regulator controller. Additionally, manoeuvres are executed with the aid of a model predictive control system.

**Figure 6 sensors-25-00117-f006:**
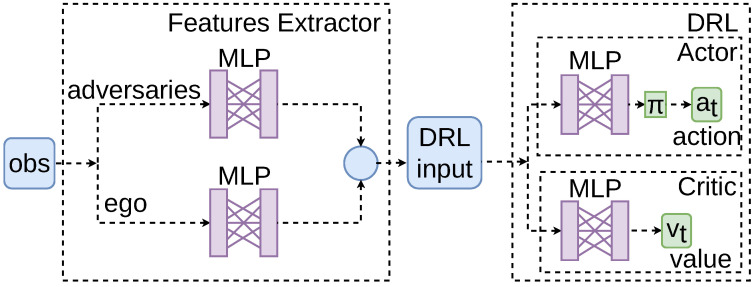
A representation of the algorithm configuration. Features from both adversaries and the ego vehicle are extracted separately. The concatenated extracted information is then input into the actor–critic structure of the DRL algorithm.

**Figure 7 sensors-25-00117-f007:**
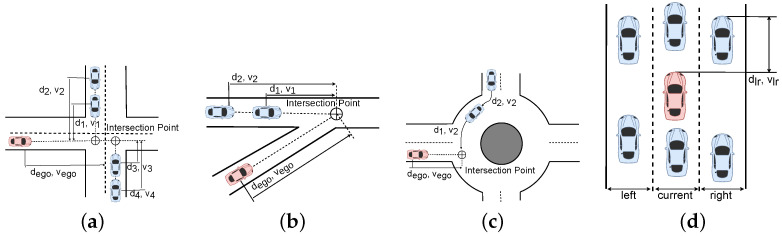
Definition of the observation matrix for each scenario. The ego vehicle (red) and the adversaries (blue) are described for each use case. (**a**) Intersection. (**b**) Merge. (**c**) Roundabout. (**d**) Lane Change.

**Figure 8 sensors-25-00117-f008:**
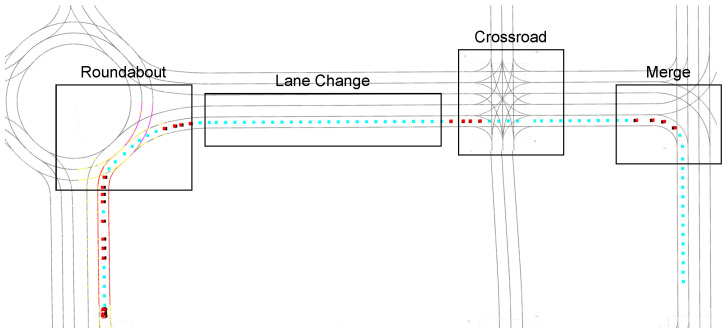
Concatenated scenario: The vehicle initially navigates through a roundabout, subsequently approaching a two-lane road populated with slow-moving vehicles. This is followed by a crossroad and, ultimately, a merge intersection. The blue dots represent the path points to be followed, while the red points signify the tactical trajectory’s use case indicators.

**Figure 9 sensors-25-00117-f009:**
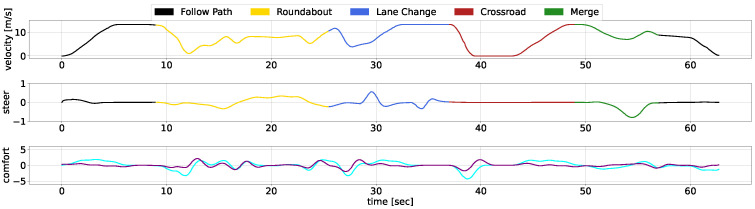
Our AD stack temporal response within the scenario. The linear velocity is depicted in the top chart, steering data are presented in the middle chart, and comfort metrics, specifically acceleration (cyan) and jerk (purple), are illustrated in the bottom chart.

**Figure 10 sensors-25-00117-f010:**
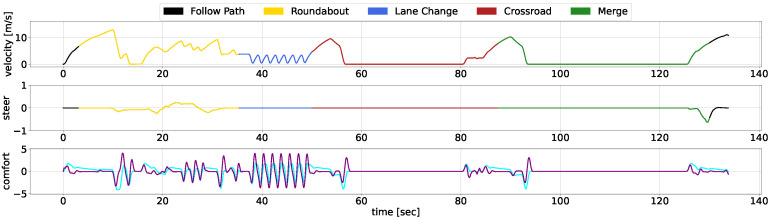
CARLA Autopilot temporal response within the scenario. The linear velocity is depicted in the top chart, steering data are presented in the middle chart, and comfort metrics, specifically acceleration (cyan) and jerk (purple), are illustrated in the bottom chart.

**Figure 11 sensors-25-00117-f011:**
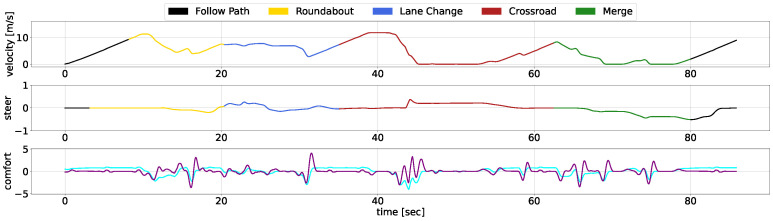
Robesafe research group architecture temporal response within the scenario. The linear velocity is depicted in the top chart, steering data are presented in the middle chart, and comfort metrics, specifically acceleration (cyan) and jerk (purple), are illustrated in the bottom chart.

**Table 1 sensors-25-00117-t001:** Comparison between the deep reinforcement learning architectures.

Architecture	Transformer-Based	Attention + LSTM	Decision Control	Tactical Behaviour	Ours
Ref.	[15]	[18]	[21]	[24]	-
State Dimensionality	High	High	Low	Low	Low
Pre-Processing	Transformer	Attention	-	-	Map
Action	Low-level	Low-level	High-level	High-level	High-level
Control Signal	Sharp	Sharp	Smooth	Smooth	Smooth
Multiple Scenario	✓	✓	×	✓	✓
Concatenated Scenario	×	×	×	✓	✓
Computational Cost	High	High	High	Low	Low
Scalability	✓	✓	×	×	✓
Real Implementation	×	✓	×	×	✓

**Table 2 sensors-25-00117-t002:** Comparison of autonomous driving system approaches.

Aspect	End-to-End	Modular	Ego-Only	Connected
Key Advantage	Reduces intermediate errors.	Enhanced interpretability and allows for parallel development.	Operates independently.	Enhanced situational awareness.
Key Disadvantage	Lacks interpretability and is difficult to implement in safety-critical systems.	Risk of error propagation and suboptimal integration of subsystems.	Limited situational awareness and no access to external data.	Dependency on reliable network infrastructure and privacy concerns.
Integration Complexity	High.	Medium.	Low.	High.
Real-World Usage	Limited.	Used.	Used.	Used.
Scalability	High.	Moderate.	High.	High.

**Table 3 sensors-25-00117-t003:** Performance metrics for the concatenated scenarios. Success rate percentage, jerk dynamics, acceleration, time to complete manoeuvres, and average speed are evaluated to assess the efficiency, smoothness, and safety of the proposals.

Metric	Ours	Autopilot	[29]
Success Rate [%] ↑	95.76	100	92.84
95th Percentile of Jerk (per episode, in m/s^3^) ↓	1.87	4.63	2.10
Maximum Jerk (per episode, in m/s^3^) ↓	3.97	4.92	4.08
95th Percentile of Acceleration (per episode, in m/s^2^) ↓	2.13	2.98	1.88
Time (s) ↓	76.85	140.23	88.43
Speed (in m/s) ↑	6.60	2.75	4.74

**Table 4 sensors-25-00117-t004:** Comparison of GT vs sensors across different weather conditions in the concatenated scenario.

Metric	Obsevations	Day	Night	Rain	Fog
Success Rate [%] ↑	GT	95.76	95.12	94.13	95.63
	Sensors	81.39	75.34	80.90	79.33

## Data Availability

No new data were created or analyzed in this study. Data sharing is not applicable to this article.

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
