# Peer review of "Enhancing Autonomous Driving in Urban Scenarios: A Hybrid Approach with Reinforcement Learning and Classical Control"

_sensors, 2024, doi:10.3390/s25010117_

Round 1
Reviewer 1 Report
Comments and Suggestions for Authors
This paper considered autonomous driving control architectures, aiming to provide a realistic implementation architecture of autonomous driving in real-world applications. Then, the proposed architecture was validated in the CARLA simulator. Although the topic is interesting and benefits the journal, this paper needs a major revision. The main concerns are as follows.
- Comments on the method and contribution.
a) The role of DRL should be detailed in the architecture, i.e., Figure 2. At the operative level, for example, we can easily figure out the roles of LQR and MPC based on Figure 5 and related descriptions. According to the descriptions in Experiments, I guess the authors want to use DRL to output ‘high-level action’.
b) The implementation of the scenario selector is unclear to me. Is it a rule-based selector? Or is it a selector obtained through learning? For example, the DRL agent automatically determines when to switch scenarios based on the observations and the learned selector.
c) I think the design of the AD architecture, as shown in Figure 2, falls somewhat into the scope of systems engineering or industrial engineering, rather than only belonging to the fields of AI and control. As a major contribution of the paper, some theoretical frameworks in systems engineering or industrial engineering may be needed to explain how the proposed architecture benefits AD performance and outperforms other architectures. Moreover, these theoretical frameworks also may enhance the quality and persuasiveness of the paper.
d) The contributions should be more accurate. For example, the combination of LQR and MPC in tracking control of AVs has been reported in the literature. The differences between the paper and existing works should be emphasized in the contributions.
e) Since the authors target on the realistic approach for driving in various complex urban scenarios, the safety should be discussed in detail in the proposed architecture, and the gap between real scenarios and simulation scenarios.
- Comments on the experiments.
a) Ablation experiments should be added to demonstrate the effectiveness of the proposed architecture. For example, DQNs are applicable to control with discrete actions and can be added, demonstrating the proposed architecture is applicable to multiple DRL algorithms.
b) The gap between real scenarios and simulation scenarios should be considered in the experiments in order to provide a realistic approach for driving in various complex urban scenarios, such as uncertainty in vehicle dynamics and somewhat different settings for the same scenario.
Reviewer 2 Report
Comments and Suggestions for Authors
The paper proposes a decision making architecture based on deep reinforcement learning (DRL) and classical control methods in autonomous driving scenarios. This paper particularly emphasizes the aspects of comfort and fast driving, which are represented by the level of jerk and the average time of episode completion. They used trust region policy optimization (TPRO) in the CARLA simulation environment, but the contribution of DRL is marginal because its use is limited to only high-level actions, such as lane change or not.
The introduction lacks a precise definition of Decision Making (DM). As DM is a broad term used across autonomous vehicle research, the paper should clarify whether it refers to steering/throttle actions, path planning, or both.
The perception module's role and input assumptions are vague. Does the architecture assume raw sensor data or pre-processed information? This needs to be explicitly detailed.
Terms such as "global trajectory" and "tactical trajectory" are not consistently aligned with the corresponding architectural levels (strategy vs. tactical). The naming conventions should reflect the stages they represent.
The forms of inputs and outputs at each stage are unclear. For instance, the representation of "perception results," "global trajectory," and "tactical trajectory" needs formal definitions to aid understanding.
Excessive use of capitalized terms (e.g., "perception," "strategy level") and abbreviations detracts from readability. Reducing such instances, except for key terms, is recommended.
Section 4 does not clearly state where DRL fits into the architecture. For example, the exact role of TRPO and its impact on tactical or strategy levels should be elaborated.
In the current implementation, DRL only decides lane change actions, while most of the control is handled at the operative level. This limits DRL's contribution to the overall system. Additionally, the experiments fail to demonstrate how much DRL improves the system's effectiveness, leaving the actual impact of DRL unproven.
In section 5.1.1, the state representation assumes access to precise positions of ego and surrounding vehicles. This assumption requires justification or discussion on its feasibility in real-world scenarios.
The details about the experimental environment, such as map configuration, vehicle specifics, and traffic density, are insufficient. A more comprehensive description would improve reproducibility.
Line 497 mentions "efficiency" but does not clarify if it pertains to runtime, computational complexity, or another factor. This ambiguity weakens the interpretation of results.
Lines 500/504 suggest the use of "limited information," but the state definition indicates substantial internal data usage. It should be clarified what kind of internal data the CARLA autopilot used, and why they are difficult to collect in real-world scenarios and are not used in the authors' approach.
Line 510 mentions that the proposed system completed the scenarios faster than the CARLA autopilot. However, the CARLA autopilot has an internal maximum speed parameter, but this aspect is not mentioned. Furthermore, are there speed limits on the road? If so, the vehicles should obey the speed limit and fast driving should be prohibited. CARLA autopilot has a parameter regarding whether the vehicle obey the speed limit or not.
The results are based on comparisons with CARLA's autopilot for a single concatenated scenario. Broader evaluations across diverse scenarios and systems would strengthen the paper's claims.
The use of linear-quadratic regulator (LQR) and model predictive control (MPC) is significant. However, a detailed discussion on their respective roles in achieving "smoothness" would enhance understanding.
Round 2
Reviewer 1 Report
Comments and Suggestions for Authors
I have no further comments.
Reviewer 2 Report
Comments and Suggestions for Authors
The authors have revised according to my previous comments.